# Determination of the Protein-Protein Interactions within Acyl Carrier Protein (MmcB)-Dependent Modifications in the Biosynthesis of Mitomycin

**DOI:** 10.3390/molecules26226791

**Published:** 2021-11-10

**Authors:** Dongjin Leng, Yong Sheng, Hengyu Wang, Jianhua Wei, Yixin Ou, Zixin Deng, Linquan Bai, Qianjin Kang

**Affiliations:** State Key Laboratory of Microbial Metabolism, Joint International Research Laboratory of Metabolic & Developmental Sciences, School of Life Sciences & Biotechnology, Shanghai Jiao Tong University, Shanghai 200240, China; dongjinleng@163.com (D.L.); John.Sheng@sjtu.edu.cn (Y.S.); wanghengyu0203@163.com (H.W.); weijianhua@sjtu.edu.cn (J.W.); yixinou@sjtu.edu.cn (Y.O.); zxdeng@sjtu.edu.cn (Z.D.); bailq@sjtu.edu.cn (L.B.)

**Keywords:** mitomycin, antitumor, acyl carrier protein, protein–protein interactions, glycosyltransferase, reductase

## Abstract

Mitomycin has a unique chemical structure and contains densely assembled functionalities with extraordinary antitumor activity. The previously proposed mitomycin C biosynthetic pathway has caused great attention to decipher the enzymatic mechanisms for assembling the pharmaceutically unprecedented chemical scaffold. Herein, we focused on the determination of acyl carrier protein (ACP)-dependent modification steps and identification of the protein–protein interactions between MmcB (ACP) with the partners in the early-stage biosynthesis of mitomycin C. Based on the initial genetic manipulation consisting of gene disruption and complementation experiments, genes *mitE*, *mmcB*, *mitB,* and *mitF* were identified as the essential functional genes in the mitomycin C biosynthesis, respectively. Further integration of biochemical analysis elucidated that MitE catalyzed CoA ligation of 3-amino-5-hydroxy-bezonic acid (AHBA), MmcB-tethered AHBA triggered the biosynthesis of mitomycin C, and both MitB and MitF were MmcB-dependent tailoring enzymes involved in the assembly of mitosane. Aiming at understanding the poorly characterized protein–protein interactions, the *in vitro* pull-down assay was carried out by monitoring MmcB individually with MitB and MitF. The observed results displayed the clear interactions between MmcB and MitB and MitF. The surface plasmon resonance (SPR) biosensor analysis further confirmed the protein–protein interactions of MmcB with MitB and MitF, respectively. Taken together, the current genetic and biochemical analysis will facilitate the investigations of the unusual enzymatic mechanisms for the structurally unique compound assembly and inspire attempts to modify the chemical scaffold of mitomycin family antibiotics.

## 1. Introduction

The mitomycins, isolated from the cultures of *Streptomyces* species [1], are an unusual class of natural product with a variety of functionalized moieties, which represent a class of antitumor agents of extraordinary potency [2,3,4]. Based on the structural and stereochemistry characteristics, the novel compact skeletons of the mitomycins were grouped into mitomycin A, B, and G types (Figure 1) [1,2,3]. The antitumor and antimicrobial activities of the mitomycins result from their alkylation of DNA by reductive activation with NADH or NADPH as the electron donor, and preference binding to the minor groove at the 5′-GC-3′ sequence [2,5]. Among of the members of the mitomycin family, the most active anticancer agent is mitomycin C, which was launched as an anticancer drug in 1974 [1,2,5]. Moreover, considerable research is still underway towards the clinical improvement of these drugs and a decrease in the side effects, such as the development of immunoconjugates with tumor-specific antibodies and high-performance drug delivery systems [6,7,8]. Because of the pharmacological potential, the unique chemical structure, and complex biosynthetic pathway, mitomycins continue to fascinate both basic and applied researchers.

The biosynthetic origins of the carbon skeleton of mitomycins were investigated by feeding ^13^C-labeled precursors followed by ^13^C-NMR analysis of the products. The results indicated that 3-amino-5-hydroxy benzoic acid (AHBA) was identified as the aromatic chromophore of mitomycin [3,9,10]. Other precursors, such as D-glucosamine, methionine, and L-citrulline, were incorporated into mitomycins for the di-pyrrole moiety, *O*-methyl groups, and a carbamate group [11,12,13,14]. The biosynthetic gene cluster of mitomycin C was disclosed from *Streptomyces lavendulae* NRRL 2564 in 1999. Forty-seven genes were identified in the 55 kb cluster, including seven genes for AHBA synthesis, together with genes putatively involved in modification, regulation, transport, and resistance [15]. A serial of gene-targeted inactivation was carried out to demonstrate the essentiality function of the corresponding genes that participated in the biosynthesis of mitomycin [15]. It was proposed that the formation of a mitosane skeleton was assembled by the CoA ligase MitE, acyl carrier protein MmcB (ACP), glycosyltransferase MitB, deacetylase MitC, reductase MitF and MitH, and the radical SAM MitD for C–C bond catalysis [2,3,15]. The defined investigations on methylation of mitomycin revealed that MitM was responsible for the N-methylation of the aziridine nitrogen [16], MitN probably catalyzed the C-9a side chain methylation of mitomycin C, and the *O*-methyltransferase MmcR methylated the 7-hydroxyl group of the aromatic moiety [15,16,17]. The carbamoylation of the C-10 hydroxyl group was identified by the carbamoyl transferase MmcS [15]. Recently, the biochemical characterization of the biosynthetic initial stage of mitosane revealed that the MitB was an ACP-dependent glycosyltransferase in mitomycin C biosynthesis [18,19].

Herein, the assembly pattern in the biosynthetic early stage of mitomycin C was initially characterized through genetic operations with *mitE*, *mmcB*, *mitB,* and *mitF* disruption and complementation, respectively. Combined with biochemical analysis and protein–protein interactions of MmcB with MitB and MitF, we dissected that the mitosane assembly underwent multiple MmcB-dependent modifications. Moreover, the combination of genetic and biochemical validations revealed that only the intact MitB display the glycosylation ability of the MmcB-tethered AHBA. A sit-directed mutagenesis experiment revealed Ser43 as the active site in MmcB for the linkage of 4’-phosphopantetheine. Remarkably, MitB and MitF exhibited close protein–protein interactions with MmcB during the mitosane assembly. These results not only provided the understanding of the interesting acyl carrier protein-dependent glycosylation and reduction steps involved in mitosane assembly, but also provided comprehensive insights into the biosynthesis of mitomycin.

## 2. Results

### 2.1. MitE Catalyzed the CoA Ligation of AHBA by the Genetic and Biochemcal Analysis 

Function annotation of *mitE* exhibited the high homology with those of the acyl-CoA synthetase-encoding genes [15]. It was considered to activate the carboxyl group of AHBA to the phosphoryl moiety by the connection of AMP, and then to transfer the AHBA-AMP substrate to CoA [15,19]. To gain genetic insight into the involvement of *mitE* in mitomycin C biosynthesis, we carried out a direct in-frame deletion of *mitE* on pJQK401 by PCR technology [20,21]. The affording Δ*mitE* recombinants, named DJ01A-C, were confirmed by PCR and sequencing analysis (Appendix A). The extracts of the validated DJ01 variants were prepared from the fermented broth and subjected to HPLC and LC-MS analysis. The results clearly displayed that the mutation of *mitE* directly disrupted mitomycin C production for the DJ01 recombinants (Figure 2). For the complementation of the mutant DJ01, the intact *mitE* was amplified and cloned under the control of the strong constitutive promoter *kasOp** into the integrated pJQK402 and introduced into DJ01 by conjugation (Figure 2). As expected, the complemented strain DJ01::*mitE* restored mitomycin C production by HPLC analysis of the culture extract. 

Encouraged by the genetic identification results, *mitE* was further heterologously expressed in *E. coli* BL21 as an *N*-terminally His_6_-tagged fusion protein. The soluble MitE was achieved and purified by Ni^2+^ affinity chromatography with the imidazole-containing elution buffer (Appendix A). The incubation of MitE with AHBA and CoA as the substrates in the presence of ATP and Mg^2+^ resulted in the appearance of AHBA-CoA by electrospray ionization-mass spectrometry (ESI-MS) analysis (Figure 3). The combination of genetic and biochemical identification not only revealed the essential role of *mitE* in mitomycin C biosynthesis, but also illuminated that the MitE was responsible for the AHBA-CoA formation in the mitomycin C biosynthetic machinery. 

### 2.2. MmcB-Tethering AHBA Triggered the Biosynthesis of Mitomycin C

The encoded product of *mmcB* was an ACP with 93 amino acid residues and executed the connection of the CoA-tethered AHBA at the active Ser site by 4’- phosphopantetheine arm [15,18,19]. We in-frame deleted *mmcB* by a mark-free strategy and generated the recombinant DJ03 by PCR screening (Appendix A). The recombinant DJ03 then fermented under the conditions known for mitomycin C production, using the wild-type strain as a positive control. The mutant DJ03 lost the production of mitomycin C, as shown by LC-MS detection (Figure 2), and that was obviously presented in the wild-type strain. The *mmcB* was further cloned into the downstream of the *kasOp** and complemented back into DJ03. The PCR-validated conjugate, DJ03::*mmcB*, recovered the mitomycin C production (Figure 2). 

The soluble *C*-terminal His_6_-tagged MmcB was achieved from *E. coli* BL21 after culture condition optimization (Appendix A). LC-MS detection revealed the occurrence of *apo*-form in the purified MmcB solution with the molecular weight as 1157.2746 ([M + H]^+^) (Figure 4A,B). After incubation of MmcB with the phosphopantetheinyl transferase (PPTase, Sfp) and CoA, the *apo*-MmcB was transferred into *holo*-form (Figure 4A,C). The conserved motif analysis and molecular homology modeling offered the 43 Ser residues as the active site for the linkage of 4’-phosphopantetheine. To identify the active Ser catalytic residue, it was generated to alanine and subsequently complement back into DJ03. The recombinant S43A MmcB abolished the ability to produce mitomycin C (Appendix A). Finally, we set out to optimize the biochemical reactions for the *apo*- and *holo*-MmcB mixture with Sfp, AHBA, ATP, CoA, and Mg^2+^. LC-MS analysis showed that the one-pot reactions were able to produce the characteristic [M + H]^+^ ions of AHBA connected with the MmcB (*m/z* = 12032.5887) through a phosphopantetheine bridge (Figure 5A). 

### 2.3. The Completed MitB Catalyzed the Glycosylation of ACP-Bound AHBA 

Further dissection of the *mitB* sequence by the FramePlot software analysis indicated the involvement of more 129 nucleotides in the upstream sequences. The BLAST analysis of the intact MitB exhibited that MitB had high sequence identity to the glycosyltransferase involved in the secondary metabolism pathways [15,18,19]. Gene *mitB* was in-frame disrupted, affording DJ05 for the identification of the involvement of *mitB* in the biosynthesis of mitomycin C (Appendix A). HPLC analysis showed that the mitomycin C production disappeared in the culture broth of DJ05 (Figure 2). Subsequently, the gene complementation of DJ05 was carried out for validation of the involvement of MitB in the mitomycin C biosynthesis with the open reading frame (ORF), the previously annotated *mitB,* and the just-refined completed one. The plasmids pJQK409 (updated) and pJQK408 (originated), carrying different *mitB* fragments, were constructed and transferred into DJ05, respectively. The recombinant strains DJ05::*mitB* (updated and originated) were individually fermented and extracted. The HPLC profile presented that the mitomycin C occurred in the DJ05::*mitB* (updated), while there was still not any rescue of mitomycin C production in DJ05::*mitB* (originated) (Figure 2). These complementation investigations examined the essential role of the *N*-terminal 43-aa residues (encoded by the 5’-terminal 129 nucleotides) for the glycosylation by MitB. 

The intact *mitB* sequence from pJQK409 was cloned into pET28a, conferring pJQK411, and transferred into *E. coli* BL21 for expression. The approximately 33 kDa molecular weight presented on the SDS-page gel of native purified MitB consisted of the calculated weight of the N-His_6_-tagged fusion protein (Appendix A). With the UDP-N-acetyl-glucosamine as the donor, the optimized one-pot reaction system containing MitB, MitE, and *holo*-MmcB in the presence of ATP, CoA, AHBA, and Mg^2+^ was established. ESI-MS analysis of the completed reaction revealed that the glycosylation product of MmcB-AHBA clearly appeared with the [M + H]^+^ ion (*m/z* = 12235.9021), referring to the MmcB-AHBA-GlcNAc (Figure 5B). This result displayed that MitB was an ACP-dependent glycosyltransferase.

### 2.4. The Protein–Protein Interactions Occurred between MmcB and MitB

ACPs play an important role, as they can establish a platform for auxiliary delivery and catalysis of related substrates in a serial of secondary metabolism biosynthesis [22,23]. Similarly, the role of ACP involvement in the mitomycin C biosynthesis is very important as it interacts with other partners to execute glycosylation and other modifications. As primary constituents of protein scaffolds and pathways, protein–protein interactions (PPIs) are crucial determinants of the loading of the building blocks during the biosynthesis of ACP template-originated natural products. In order to establish the interaction between MmcB and MitB, tag-free MmcB and MitB were also achieved from *E. coli* BL21, respectively (Appendix A). The initial pull-down assay indicated that MmcB and MitB have an obvious protein–protein interaction. Subsequently, the affinity for the interaction of MmcB with MitB was estimated by SPR biosensor analysis. During the detection, 5000 RU of MitB as a probe molecule was immobilized on a CM5 chip, then the targeted MmcB solutions with the different concentrations of MmcB were successively flown into contact with the surface of MitB. The supplement of 2.7–43.2 μM MmcB solutions induced a gradually increased affinity interaction of the MitB-bound surface, displaying a stable binding between MmcB and MitB. The results of the SPR evaluation of binding kinetics revealed that the association rate constant (*K*_a_) and dissociation rate constant (*K*_d_) were 1.95 × 10^5^ M^−1^S^−1^ and 0.113 s^−1^, respectively (Figure 5C). The equilibrium dissociation value *K_D_* was 5.78 × 10^−7^ M, and the binding ratio of MmcB and MitB was 1:1. 

### 2.5. MmcB Interacted with MitF during Mitomycin C Assembly

The gene *mitF* encoded a protein belonging to the family of reductase [2,15], which was predicted involved in the early-stage biosynthesis of mitomycin C. To further correlate the function of *mitF*, the detailed phylogenetic analysis revealed that MitF was classified into the clade containing of ACP-dependent reductases (Appendix A). Further homology modeling of MitF with FabG from *E. coli* indicated that these reductases evolved to similar active centers and key residues responsible for ACP interactions *via* the SWISS-MODEL server [24,25] (Appendix A). In order to explore its proposed reactions, *mitF* of the strain NRRL 2564 was cloned into the pET30a protein expression vector, conferring pJQK414, and transferred into *E. coli* BL21(DE3) for heterologous expression. SDS-PAGE analysis of the purified protein clearly exhibited the production of the soluble N-terminal His_6_-tag-fused MitF (Appendix A). Then, the interaction of MitF with MmcB was evaluated by pull-down assay (Appendix A) and SPR biosensor analysis according to the protocol. Similarly, 3000 RU MitF was fastened on a CM5 chip as the probe molecule, and 0.7–21.6 μM MmcB was continuously supplemented with MitF. The affinity analysis results indicated that the association rate constant (*K*_a_) and dissociation rate constant (*K*_d_) were 5.58 × 10^3^ M^−1^S^−1^ and 0.061 s^−1^, respectively (Figure 6). The equilibrium dissociation value *K_D_* was 1.08 × 10^−7^ M, and the binding ratio of MmcB and MitF was 1:1. 

To determine the involvement of *mitF* in the biosynthesis of mitomycin C, the targeted inactivation of mitF was carried out through in-frame gene deletion and generated the mutant DJ08. The cultures of the LDJ08 were extract and subjected to the HPLC analysis. As shown in Figure 2, the resultant mutant LDJ08 completely lost the ability to produce mitomycin C. Subsequently, *mitF* was cloned under the control of the *kasOp** for gene complementation, generating the recombinant strain LDJ08::*mitF*. HPLC-MS analysis of the extract of LDJ08::*mitF* showed a chromatogram identical to that of the wild-type strain. The production of mitomyin C was fully restored by complementation of *mitF* (Figure 2). These investigations identified that *mitF* was essential in the biosynthesis of mitomycin C.

## 3. Discussion

The antitumor agent MC was first discovered from the fermented broth of *S. lavendulae* NRRL2564 and has been used in the treatment of a variety of tumors. Its natural derivatives with varying biological activities and chemical structures have been identified from the cultures of different Actinobateria [2,3,26,27]. The previous feeding experiments with isotope-labeled precursors established that AHBA, D-glucosamine, carbamoyl phosphate, and S-adenosyl methionine were the basic building blocks for biosynthetic assembly of mitomycin C analogs [2,3,11,14]. Remarkably, AHBA was an important precursor for a large group of natural products, including the family of naphthalenic and benzenic ansamycins, the unique saliniketals, and the family of mitomycins [3]. The adenylation of AHBA catalyzed by the A domain anchored at the loading domain initiated the PKS assembly in ansamycin or saliniketal biosynthesis. In the mitomycin biosynthesis, the activation of AHBA *via* a two-step reaction of adenylation followed by thioesterification was catalyzed by the CoA ligase (MitE), according our current results and the recent investigations [2,3,18,19]. Moreover, we show that the CoA activation of 3,5-AHBA and loading onto MmcB by MitE initiated the biosynthesis of mitomycin. 

In our research, the molecular genetic manipulation and biochemical analysis mutually identified the essential role of the involvement of functional genes and encoded proteins in the biosynthesis of the early stage of mitomycin C. The refined gene inactivation and complementation manipulations of *mitE*, *mmcB*, *mitB,* and *mitF* revealed their essential role and involvement in the biosynthesis of the early stage of mitomycin C (Figure 2). The gene complementation experiment of *mitB* efficiently revealed the necessary functional region of the 129 nucleotides that are located upstream of the original start codon [28]. Moreover, biochemical analysis characterized that MitB was an ACP-dependent glycosyltransferase. MmcB and MitB displayed a clear relationship with a binding ratio of 1:1 by SPR biosensor analysis, with the equilibrium dissociation value *K_D_* of 5.78 × 10^−7^ M. Importantly, the protein–protein interaction estimation also showed that MmcB was in relation to MitF, with the equilibrium dissociation value *K_D_* of 1.08 × 10^−7^ M, and the binding ratio of 1:1. Encouraged by the SPR analysis, genetic validations of the *mitF* were carried out and the results exhibited its indispensable role in the mitomycin C biosynthesis. This function characterization results of MitF implied that it was an ACP-dependent catalysis pattern as well. 

Remarkably, the ACP-dependent tailoring enzymes are involving in the biosynthesis of a large number of important chemical molecules, from primary metabolites to structurally complex natural products [18,29,30]. The ACP is a relatively small protein with a dynamic and monomeric helical bundle architecture, which shuttles the nascent cargo to the partner enzymes for decoration through the biosynthetic machinery. In the biosynthesis of natural product, the nascent intermediate is covalently tethered to the *holo*-ACP *via* thioester linkage to a post-translationally substituted phosphopantetheine (PPant) arm, which is activated by PPTase from the *apo*-ACP [29,30]. Moreover, the ACP protects the growing nascent intermediates from non-selective modification or degradation in the cytosol by sequestering them within a central hydrophobic core [31]. During the continuously maturation of the intermediate, ACP must form specific protein–protein interactions, and the tethered intermediate should be translocation from the hydrophobic core of ACP into the active site of the partners. The ACP-mediated assembly pattern in the natural product biosynthesis pathway offers a safe, efficient, and economical program to facilitate the direct production of the final compounds [22,30,32]. Combined with our present investigations, the AHBA-tethered MmcB triggered the biosynthesis of the early-stage of mitomycin C. In this reaction, MmcB specifically interacted with the partners, such as MitE, MitF, and others, to afford the mitosane skeleton. In addition, based on the MmcB interactions analysis with the partners, the other under explored proteins involved in the mitosane assembly might be determined. 

Finally, a complete understanding of the biosynthesis of the antitumor agent mitomycin C not only provides a foundation to investigate the enzymatic mechanisms for assembling the structurally unique and pharmaceutically important groups, but also opens an opportunity for further engineering to improve the bioactivities of these compounds.

## 4. Materials and Methods

### 4.1. Bacterial Strains, Plasmids, and Culture Conditions

The strains, plasmids, and primers used in this study are listed in Appendix A. Strain NRRL2564, the wild-type producer for mitomycin C, was obtained from Agricultural Research Service Culture Collection. 

*Escherichia coli* strains were grown in LB broth or agar (10 g/L tryptone, 5 g/L yeast extract, 5 g/L NaCl, or 20 g/L agar) at 37 °C. Strain NRRL2564 and its mutants were grown in TSBY liquid medium (30 g/L tryptic soy broth, 103 g/L sucrose, 5 g/L yeast extract, pH 7.2) for collection of mycelium or on ISP2 solid medium (4 g/L yeast extract, 10 g/L malt extract, 4 g/L glucose, 20 g/L agar, pH 7.2) at 30 °C for spore formation. The conjugation of Streptomyces and *E. coli* was carried out on SFM agar plates (20 g/L soybean powder, 20 g/L mannitol, 20 g/L agar, pH 7.2). 

### 4.2. Gene Deletion and Complementation 

#### 4.2.1. Gene MitE In-Frame Deletion and Complementation

The targeted in-frame deletion of *mitE* was carried out in a homologous recombination strategy. The two 1.5 kb fragments of homologous arms located upstream and downstream of the deleted region were amplified from the gDNA of strain NRRL2564 with the two pairs of primers (*mitE*-Left-F/R and *mitE*-Right-F/R), respectively. Confirmed by DNA sequencing, the upstream fragment was digested by *Kpn*I-*Hin*dIII, the downstream fragment was cut by *Hin*dIII-*Bam*HI, and the two fragments were then cloned into the *Kpn*I-*Bam*HI sites of pJTU1278 to give pJQK401. The pJQK401 was transferred into *E. coli* ET12567/pUZ8002 and further introduced into strain NRRL2564 by *E. coli*-*Streptomyces* biparental conjugation [15,21]. The exconjugants were selected for thiostrepton resistance (25 μg/mL), and the double-crossover mutants were obtained after two rounds of no-selective growth. The validation of the mutant DJ01 through PCR generated an expected PCR fragment corresponding to 0.52 kb, whereas the fragment produced from the wild-type strain was 2.2 kb (Appendix A). 

To complement the *mitE* disruption mutant DJ01, gene *mitE* was amplified from the gDNA of strain NRRL2564 with the primer of mitEHB-F/R. The sequencing validated *mitE* (digested with *N**de*I&*E**co*RI) was cloned into the pSET152-derived plasmid (digested with *N**de*I&*E**co*RI), and under the control of *kasOp** it afforded pJQK402. Then, pJQK402 was transferred into mutant DJ01 by conjugation from *E*. *coli* ET12567/pUZ8002. The apramycin-resistant recombinants were selected from the initial exconjugants. The recombinants were further verified by PCR amplification with primers mitEHB-F&-R. Finally, the culture broth of the corrected candidates was prepared and subjected to HPLC analysis. 

#### 4.2.2. Gene MmcB In-Frame Deletion and Complementation

The homologous DNA fragments situated upstream and downstream of the in-frame deletion region of *mmcB* were amplified by PCR and inserted into pSK (+), respectively. Confirmed by DNA sequencing, the upstream fragment was digested by *Kpn*I-*Hin*dIII, and the downstream fragment was cut by *Hin*dIII-*Bam*HI. Then, the two fragments were cloned into *Kpn*I-*Bam*HI sites of pJTU1278 to form pJQK404. The pJQK404 was transferred into *E. coli* ET12567/pUZ8002 for conjugation with strain NRRL2564. The thiostrepton-resistant recombinants were selected for two rounds of no-selective growth. The double-crossover mutants were obtained by PCR screening with primers mmcB-F/R, named DJ03. Instead of a 135 bp fragment from the wild-type strain, the PCR product from the mutant was 520 bp, whereas the PCR amplification fragment was 655 bp from the wild-type strain (Appendix A). Three single clones of each recombinant were independently used for LC-MS analysis of the production of mitomycin C.

For the complementation of gene *mmcB* back into mutant DJ03, the promoter *kasOp** was introduced upstream of this gene in the pSET152-originated plasmid. Gene *mmcB* was amplified with the primers of mmcBHB-F/R. Then, the sequenced fragment was cloned into the *N**de*I&*E**co*RI double digested plasmid pSET152, resulting in the construction of plasmid pJQK405. The recombinant plasmid was then transferred *via E. coli* ET12567/pUZ8002 into the *mmcB*-mutant strain DJ03 by intergeneric conjugation for generating strain DJ03::*mmcB* with pJQK405. Three single clones of each recombinant were independently used for LC-MS analysis of the production of mitomycin C.

#### 4.2.3. Gene MitB In-Frame Deletion and Complementation

The homologous DNA fragments situated upstream and downstream of the in-frame deletion region of *mitE* were amplified by PCR and inserted into pSK (+), respectively. Confirmed by DNA sequencing, the upstream fragment was digested by *Kpn*I-*Hin*dIII, and the downstream fragment was cut by *Hin*dIII-*Bam*HI. Then, the two fragments were cloned into *Kpn*I-*Bam*HI sites of pJTU1278 to form pJQK407. The pJQK407 was transferred into *E. coli* ET12567/pUZ8002 for conjugation with strain NRRL2564. The thiostrepton-resistant recombinants were selected for two rounds of no-selective growth. The double-crossover mutants were obtained by PCR screening with primers mitB-F/R, named DJ05. Instead of a 477 bp fragment from the wild-type strain, the PCR product from the mutant was 176 bp, while the obtained PCR product was 641 bp from the wild-type stain (Appendix A). Three single clones of each recombinant were independently used for LC-MS analysis of the production of mitomycin C.

For the complementation of gene *mitB* back into mutant DJ05, the promoter *kasOp** was introduced upstream of this gene in the pSET152-originated plasmid. Gene *mitB* was amplified with the primers of mitBHBO-F/R and mitBHBU-F/R. Then, the sequenced fragment was cloned into the *N**de*I&*E**co*RI double digested plasmid pSET152, resulting in the construction of plasmid pJQK408 and pJQK409, respectively. The recombinant plasmids were then transferred *via E. coli* ET12567/pUZ8002 into the *mitB*-mutant strain DJ05 by intergeneric conjugation for generating strain DJ05::*mitB* (originated) and DJ05::*mitB* (updated) with pJQK408 and pJQK409, respectively. Three single clones of each recombinant were independently used for LC-MS analysis of the production of mitomycin C.

#### 4.2.4. Gene MitF In-Frame Deletion and Complementation

The homologous DNA fragments situated upstream and downstream of the in-frame deletion region of *mitF* were amplified by PCR and inserted into pSK (+), respectively. Confirmed by DNA sequencing, the upstream fragment was digested by *Bam*HI-*Hin*dIII, and the downstream fragment was cut by *Hin*dIII-*Kpn*I. Then, the two fragments were cloned into *Kpn*I-*Bam*HI sites of pJTU1278 to form pJQK412. The pJQK412 was transferred into *E. coli* ET12567/pUZ8002 for conjugation with strain NRRL2564. The thiostrepton-resistant recombinants were selected for two rounds of no-selective growth. The double-crossover mutants were obtained by PCR screening with primers mitF-F/R, named DJ08. Instead of a 906 bp fragment from the wild-type strain, the PCR product from the mutant was 510 bp (Appendix A). Three single clones of each recombinant were independently used for LC-MS analysis of the production of mitomycin C.

For the complementation of gene *mitF* back into mutant DJ08, the promoter *kasOp** was introduced upstream of this gene in the pSET152-originated plasmid. Gene *mitF* was amplified with the primers of mitF-F/R. Then, the sequenced fragment was cloned into the *N**de*I&*E**co*RI double digested plasmid pSET152, resulting in the construction of plasmid pJQK413. The recombinant plasmid was then transferred *via E. coli* ET12567/pUZ8002 into the *mitF*-mutant strain DJ08 by intergeneric conjugation for generating strain DJ08::*mitF* with pJQK509. Three single clones of each recombinant were independently used for LC-MS analysis of the production of mitomycin C.

### 4.3. Production and Detection of Mitomycin C in Strain NRRL2564

For fermentation, strain NRRL2564 and its mutants were grown in the seed medium TSBY for 24 h at 30 °C; then, 4% inoculum was transferred into the fermentation medium GSY (15 g/L glucose, 5 g/L soluble starch, 5 g/L yeast extract, 5 g/L NaCl, 5 g/L CaCO_3_, pH 7.2) and cultured for 4 days at 220 rpm, 30 °C. 

To detection the production of mitomycin C in the wild-type strain NRRL2564, the mutants, and the complemented recombinants, 50 mL of fermentation broth was extracted with 50 mL of ethyl acetate. After vacuum evaporated, the crude extract was dissolved in 1 mL methanol and passed through 0.22 μm filters, and prepared for HPLC (Agilent series 1260, Agilent Technologies, Santa Clara, CA, USA) analysis. The HPLC was monitored under the following conditions: column, Agilent Eclipse SB-C18 column (250 × 4.6 mm, 5 μm); mobile phase, 0.1% formic acid in water (solvent A) and acetonitrile (solvent B); flow rate: 0.6 mL/min; gradient condition, 85% A: 15% B for 5 min, 80% A: 20% B at 6 min, 40% A: 60% B at 30 min, 10% A: 90% B at 31 min, and after 5 min, the conditions returned back to 85% A: 15% B for 5 min; injection volume, 10 μL; and column temperature, 37 °C, UV 363 nm.

### 4.4. Protein Expression and Purification 

Genes *mitE*, *mitB,* and *mitF* were amplified from the gDNA of strain NRRL2564 with the corresponding primers (Appendix A), and cloned into the *Nde*I and *Eco*RI sites of the pET28a vector for sequencing. The corrected plasmids were then transferred into *E. coli* BL21 (DE3). Gene *mmcB* was synthesis with the optimized codon for *E. coli* heterologous expression and cloned into pET28a to produce the related plasmids. The corresponding plasmids were then transferred into *E. coli* BL21 (DE3). The recombinant strains were then individually inoculated into 5 mL LB medium containing 50 μg/mL kanamycin and incubated at 220 rpm, 37 °C overnight. Then, 1 mL of seed cultures were transferred into 500 mL flasks containing 100 mL fresh LB medium incubated at 37 °C until the OD_600_ reached at about 0.6, 0.5 mM IPTG was supplied, and then they kept incubating for another 20 h at 18 °C. 

To purify the His_6_-tagged recombinant proteins, cells were harvested by centrifugation at 5000 rpm, 4 °C for 10 min, and then the pellet was resuspended in 40 mL binding buffer (500 mM NaCl, 20 mM Tris, 5 mM imidazole, pH 7.9). After sonication on ice for 20 min for 20 s at 20 s intervals, the cell debris was removed by centrifugation at 12,000 rpm and kept at 4 °C for 30 min. The targeted protein was purified from the soluble cell extract by Ni-NTA column (GE Healthcare, Ni Sepharose 6 Fast Flow) according to the manufacturer’s instructions. The fractions were eluted by buffer A containing 60 mM, 80 mM, 100 mM, or 250 mM imidazole, up to 100 mL in total, and concentrated to 1 mL in buffer A in a 30 kD Amicon Ultra-15 Centrifugal Filter (Millipore, Burlington, MA, USA). Finally, SDS-PAGE was carried out and protein bands were stained with Coomassie Brilliant Blue R250. Protein concentrations were determined by nano-drop (Appendix A). For the analysis of protein–protein interactions by protein pull-down and SPR assays (Figure 4 and Figure 5*,*
Appendix A), the purified protein was concentrated using nickel-NTA chromatograph and subjected into fast protein liquid chromatography (FPLC) for further purification. The highly purified proteins (purity > 92%) were used for analysis or stored at −80 °C.

### 4.5. Enzyme Activity Analysis 

The MitE assay reaction contained Tris-HCl (pH 7.5) 100 mM, MgCl_2_ 2.0 mM, DTT 1 mM, ATP 1 mM, AHBA 1 Mm, CoA 1 mM, and purified 4 µM MitE in a final volume 200 µL. The reactions were carried out at 37 °C for 30 min and terminated by flash frozen at −80 °C. Then, the mixture was subjected into LC-MS analysis on an Agilent 1290 Infinity II/6545 QTOF LC/MS instrument (Agilent Corp., Santa Clara, CA, USA), with an Agilent Eclipse SB-C18 column (250 × 4.6 mm, 5 μm). 

The soluble expression of *holo*-MmcB was performed in *E. coli* BL21(DE3)/pSV20, which contains the Sfp for the *in vivo* phosphopantetheinylation of ACP. Meanwhile, *E. coli* BL21(DE3)/pSV20 is used for the expression and purification of the *holo*-ACP at the Ser active site, as previously described [33]. For the formation of the MmcB-tethered AHBA, the purified *holo*-MmcB (5 µM), in a volume of 200 µL, was supplement in the MitE reaction mixture as described above and incubated for 1 h at 37 °C. The reactions were quenched by flash frozen at –80 °C and analyzed with QTOF LC/MS instrument after centrifugation for 10 min at 12,000 rpm and filtration.

The complete one-pot reaction 200 µL system, including MitE (4 µM) and holo-MmcB (5 µM), in the presence of Tris-HCl (pH 7.5) 100 mM, ATP 1 mM, CoA 1 mM, AHBA 1 mM, DTT 1 mM, and MgCl_2_ 2.0 mM, was initiated by adding MitB (4 µM) and UDP-*N*-acetyl-glucosamine (5 µM) and then incubated for 2 h at 37 °C. Finally, the reaction was quenched by flash freezing at −80 °C. The resultant aqueous solution was centrifuged and subjected to QTOF-LC/MS instrument analysis. 

### 4.6. Determination of the Protein Molecular Weight by Q-TOF Mass Spectrometry

To detect the molecular weight of apo-ACP, holo-ACP, and different ACP-tethered substrates or products, the mass spectrometry assays were monitored on a 6530 Accurate-Mass Q-TOF spectrometer coupled with an Agilent HPLC 1260 series using a 5% to 95% linear gradient of acetonitrile/water at a flow rate of 0.4 mL/min. Data were obtained in the positive mode and the mass scan range was set between 600 and 2500 *m/z*. The resulting spectrums were analyzed using the Mass Hunter to calculate the masses of the intact proteins.

### 4.7. Protein–Protein Interaction Analysis

The C-terminal His_6_-tagged *holo*-MmcB was separated from the soluble cell lysate of *E. coli* BL21(DE3)/pSV20 by metal affinity chromatography, in which the His_6_-tagged *holo*-MmcB was situated to the downstream of enterokinase digestion site. The His_6_-tagged *holo*-MmcB was digested by the recombinant enterokinase according to the manufacturer’s instructions and further purified by His-bind resin for elimination of the His_6_-tagged peptides. Subsequently, the protein pull-down assays between MitB or MitF and tag-free *holo*-MmcB were performed. The protein pull-down analysis between MitB and MitF was carried out in a total volume of 100 µL in 10 Mm PBS buffer (pH 7.5) and incubated for 1 h at 30 °C. Then, the reaction mixtures were submitted to His-bind resin chromatography, and the fractions were eluted by PBS buffer containing 20 mM, 40 mM, 60 mM, 80 mM, or 100 mM imidazole, up to 100 mL in total, and concentrated to 500 µL in PBS buffer in a 10 kD Amicon Ultra-15 Centrifugal Filter (Millipore) for SDS-PAGE analysis.

The protein–protein dissociation constants between MitB or MitF and *holo*-MmcB were estimated by surface plasmon resonance biosensor analysis using a Biacore 8K biosensor (GE Healthcare Life Sciences, USA) at 25 °C. For determination of the protein–protein interactions, the purified MitB was bound to the Series S sensor chip CM5. In detail, 20 µg/mL MitB or MitF in 10 mM NaAc (pH 4.0) was coupled to the CM5 chip at 5000 and 3000 RU, respectively, and then different *holo*-MmcB solutions were flown through the chip, and the signal responses were recorded. The signal of the solution without *holo*-MmcB was used as a control, and the value was subtracted from other values during data processing (Figure 4 and Figure 5).

## 5. Conclusions

In summary, the gene disruption investigations revealed the requisites of *mitE*, *mmcB*, *mitB,* and *mitF* during mitomycin biosynthesis, and about whichever gene disruption will destroy the production of mitomycin C in the recombinant strains. Based on the biochemical analysis, we determined that MitE was responsible for transferring CoA to 3-amino-5-hydroxy-bezonic acid (AHBA), and was then loaded onto MmcB for the initiation of the biosynthesis of mitomycin C. The in vitro protein–protein interaction analysis with the pull-down assay and SPR biosensor detection defined that MmcB interacted with MitB and MitF, and the equal proportion binding status was defined as 1:1. This result indicated that both MitB and MitF were ACP (MmcB)-dependent modification enzymes in mitomycin biosynthesis. Finally, the current genetic and biochemical investigations enriched our understanding in the early stage of mitomycin C assembly and will encourage the investigations on the novel enzymatic mechanism for the building of structurally unique mitomycins. 

## Figures and Tables

**Figure 1 molecules-26-06791-f001:**
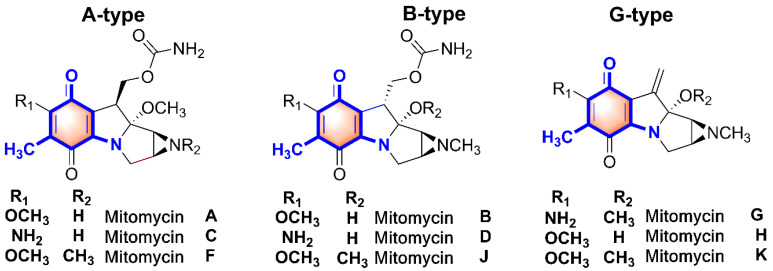
The varied chemical structures of mitomycins.

**Figure 2 molecules-26-06791-f002:**
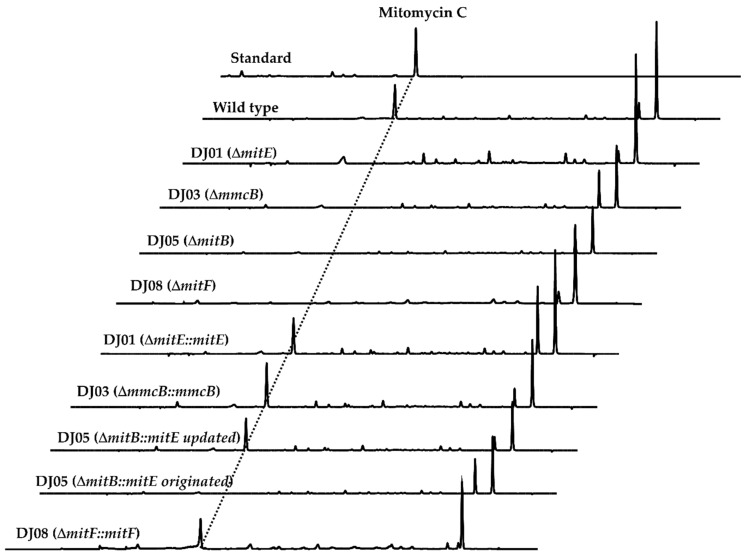
HPLC analysis results of the genetic recombinant strains.

**Figure 3 molecules-26-06791-f003:**
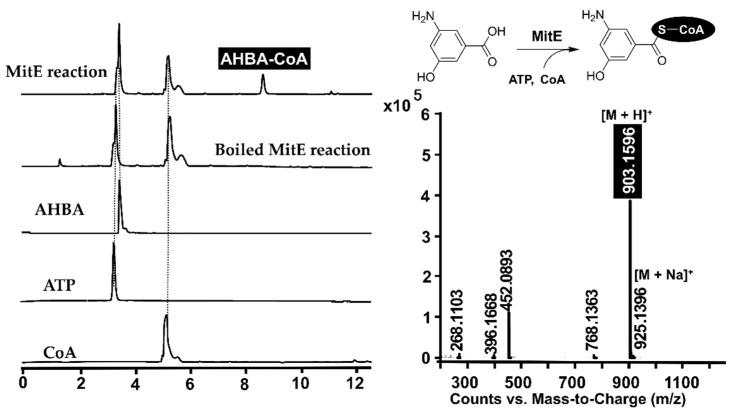
The HPLC-MS analysis of the AHBA-CoA formation.

**Figure 4 molecules-26-06791-f004:**
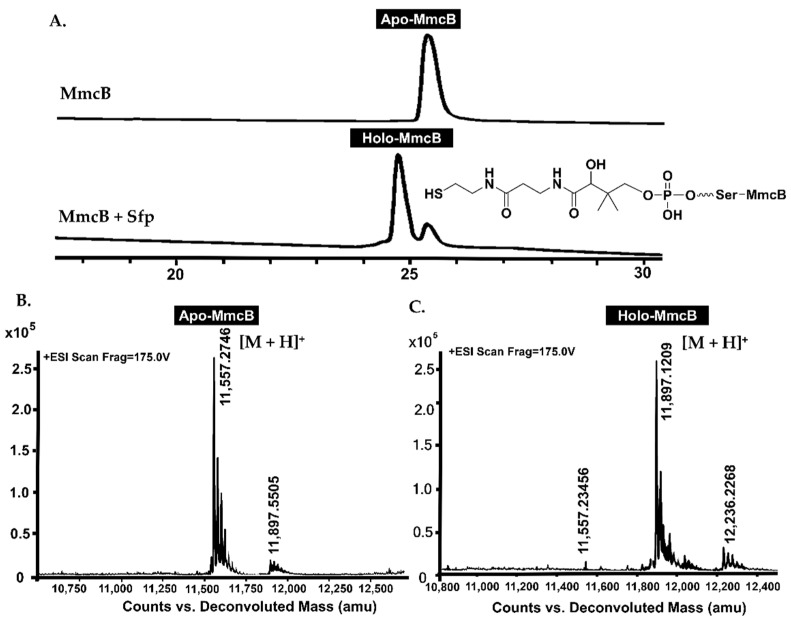
The LC-MS analysis results of the *apo*- and *holo*-form of MmcB. (**A**) The retention time of the *apo*- and *holo*-form of MmcB by HPLC analysis. (**B**) The molecular weight of the *apo*-form MmcB. (**C**) The molecular weight of the *holo*-form MmcB.

**Figure 5 molecules-26-06791-f005:**
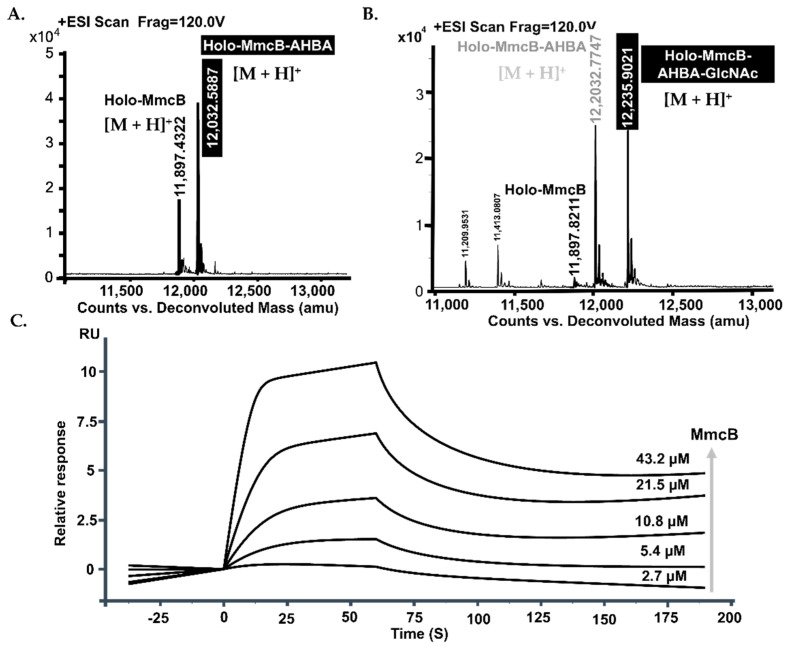
MitB catalyzed the glycosylation of MmcB-tethered AHBA. (**A**) ESI-MS analysis of the formation of MmcB-AHBA. (**B**) ESI-MS analysis of the formation of MmcB-AHBA-GlcNA. (**C**) The determination of the interaction between MitB and MmcB by Biacore biosensor assay.

**Figure 6 molecules-26-06791-f006:**
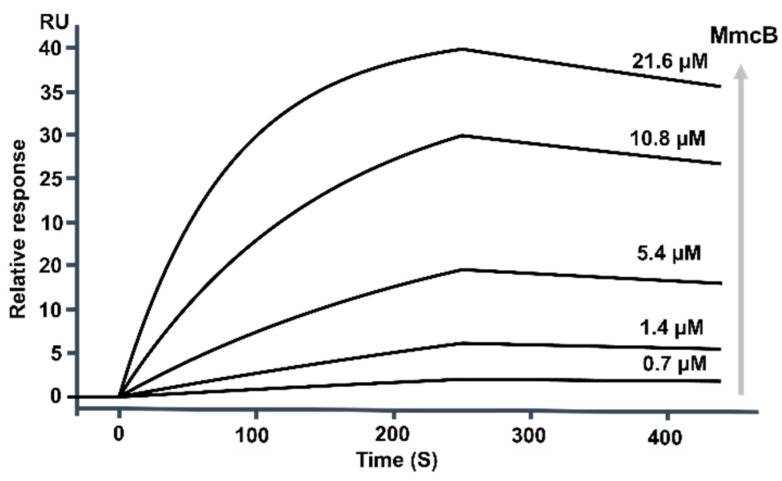
The determination of the interaction between MitF and MmcB by Biacore biosensor assay.

## Data Availability

The data presented in this study are available on request from the corresponding author.

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
