# Peer review of "Determination of the Protein-Protein Interactions within Acyl Carrier Protein (MmcB)-Dependent Modifications in the Biosynthesis of Mitomycin"

_molecules, 2021, doi:10.3390/molecules26226791_

Round 1
Reviewer 1 Report
The manuscript describe the study involved gene disruption of MitE, MmcB, MitB and MitF to determine genes required for mitomycin biosynthesis. In addition, in vitro assay was carried out to test the hypothesis that CoA was transferred to AHBA by MitE, followed by MmcB was interacted at the beginning of mitomycin C production. In conclusion, Authors found that MitB and MitF were involved in ACP dependent modifications by MmcB. This conclusion was further confirmed by in vitro pull down assay SPR biosensor analysis.
1) Page 7, line 233, D-glucosamine should be D-glucosamine.
2) Page 1, line 36, Streptomyces should be Streptomyces
3) Page 3, line 105, N-terminally should be N-terminally.
4) Supplementary materials, Fig. S10, I suppose MitF protein structure was constructed using FabG as template, please provide the PDB id code.
Author Response
Part I (reviewer 1):
Major points
The manuscript describe the study involved gene disruption of mitE, mmcB, mitB and mitF to determine genes required for mitomycin biosynthesis. In addition, in vitro assay was carried out to test the hypothesis that CoA was transferred to AHBA by MitE, followed by MmcB was interacted at the beginning of mitomycin C production. In conclusion, Authors found that MitB and MitF were involved in ACP dependent modifications by MmcB. This conclusion was further confirmed by in vitro pull down assay SPR biosensor analysis.
1) Page 7, line 233, D-glucosamine should be D-glucosamine.
2) Page 1, line 36, Streptomyces should be Streptomyces
3) Page 3, line 105, N-terminally should be N-terminally.
4) Supplementary materials, Fig. S10, I suppose MitF protein structure was constructed using FabG as template, please provide the PDB id code.
The answer: Thanks for the carefully review and kindly suggestion. We have corrected all of the improper presentation. The PDB id code of FabG has offered at Figure Sx.

Reviewer 2 Report
The manuscript of Dongjin Leng et al. is well written and informative. The Experimental design and results are sound and interesting. However, the authors need to first summarize the major conclusions of the paper and then move on to addressing the potential importance of these conclusions.
Minor points:
There are some word choice/grammatical errors throughout the manuscript
Author Response
The manuscript of Dongjin Leng et al. is well written and informative. The Experimental design and results are sound and interesting. However, the authors need to first summarize the major conclusions of the paper and then move on to addressing the potential importance of these conclusions.
There are some word choice/grammatical errors throughout the manuscript
The answer: We are greatly thanks for the kindly comments and suggestions. We have revised all of the errors as we can in the manuscript.

Reviewer 3 Report
Leng et al, carry out a study that helps to understand the biosynthesis of a very important biomolecule in terms of its applications. The present study advances in the understanding of the interactions of the MmcB protein with MitB and MitF.
I have some considerations on some points that I ask the authors to clarify for me.
Major revisions:
Line 96: Figure S1-B, I do not know if it was perhaps writing wild instead of wide type. Where is the negative control?
Line 99: Figure 2. Why was it not analyzed in the HPLC to the purified mitomycin molecule to be able to compare the peaks of the signal?
Line 119: Figure S2-B, I do not know if it was perhaps writing wild instead of wide type. Where is the negative control?
Line 149: Figure S4-B, I do not know if it was perhaps writing wild instead of wide type. Where is the negative control?Line 199: Why was the construction of the phylogenetic tree not described in the materials and methods, or in this section.
Line 200: The sequences that were used for tree construction all have ACP-dependent reductase functional annotation, so the assertion that MitF falls into the reductase clade is redundant and obvious.
Minor revisions:
Line 90: Functional notations can not have homology between them. Homology is an evolutionary concept.
Line 110: Delete "of" before "not only" .
Line 120: Did you mean wild type strain?
Line 122: Did you mean wild type strain?
Line 145-146: Results obtained with BLAST are in terms of sequence identity, not homology. I consider that the idea to express is that the amino acid sequence of MitB has a greater identity with the amino acid sequence of glycosyltransferase. In case the authors have determined homology relationships between the sequences, they should explain how they did it.
Line 179: Delete "of" before "between".
Line 200: Figure S9, Perhaps the authors mean "Phylogenetic analysis of MitF with its homologues" instead of "Phylogenetic analysis of MitF with the homologies".
Line 208: Did you mean protocol?
Author Response
Leng et al, carry out a study that helps to understand the biosynthesis of a very important biomolecule in terms of its applications. The present study advances in the understanding of the interactions of the MmcB protein with MitB and MitF.
I have some considerations on some points that I ask the authors to clarify for me.
Major revisions:
Line 96: Figure S1-B, I do not know if it was perhaps writing wild instead of wide type. Where is the negative control?
Line 119: Figure S2-B, I do not know if it was perhaps writing wild instead of wide type. Where is the negative control?
Line 149: Figure S4-B, I do not know if it was perhaps writing wild instead of wide type. Where is the negative control?
The answer: We thank the reviewer for pointing out these issues and providing very kindly comments. As suggested, we are sorry for the mistakes and the “wide type” have been replaced by “wild type” in the manuscript. Moreover, the no template-added PCR reactions were the negative control.
Line 99: Figure 2. Why was it not analyzed in the HPLC to the purified mitomycin molecule to be able to compare the peaks of the signal?
The answer: This is a very good suggestion. The standard of purified mitomycin C was supplemented in the HPLC analysis profile.
Line 199: Why was the construction of the phylogenetic tree not described in the materials and methods, or in this section.
The answer: Thanks for the kindly suggestion. The description for the construction of the phylogenetic tree had provided in the figure legend.
Line 200: The sequences that were used for tree construction all have ACP-dependent reductase functional annotation, so the assertion that MitF falls into the reductase clade is redundant and obvious.
The answer: Thanks for the good suggestion, the redundant expression had been deleted in the current manuscript.
Minor revisions:
Line 90: Functional notations can not have homology between them. Homology is an evolutionary concept.
Line 110: Delete "of" before "not only" .
Line 120: Did you mean wild type strain?
Line 122: Did you mean wild type strain?
Line 179: Delete "of" before "between".
Line 200: Figure S9, Perhaps the authors mean "Phylogenetic analysis of MitF with its homologues" instead of "Phylogenetic analysis of MitF with the homologies".
Line 208: Did you mean protocol?
The answer: We are greatly thanks for the carefully reviews. All of the comments and suggestions greatly improved our manuscript, and all of the inappropriate presentations have been revised.
Line 145-146: Results obtained with BLAST are in terms of sequence identity, not homology. I consider that the idea to express is that the amino acid sequence of MitB has a greater identity with the amino acid sequence of glycosyltransferase. In case the authors have determined homology relationships between the sequences, they should explain how they did it.
The answer: It is really a good suggestion and we have revised the explanation. Moreover, we performed the sequence-based searches against the database of non-redundant protein sequences by Blastp program in NCBI server. The result showed that MitB displayed higher sequence identity with the reported family 2 glycosyltransferases (Figure S4).

Reviewer 4 Report
The paper entitled “Determination of the Protein-Protein Interactions Within Acyl 2
Carrier Protein (MmcB) Dependent Modifications in the Bio- 3
synthesis of Mitomycin” is a very interesting paper focused on a crucial topic.
But several issues should be addressed before publication:
1)the scientific message of this study resulted a bit obscure and also the last part should be re-written
2) in the conclusion section where the authors write “pave the way towards the attempts to modify the chemical structures of mitomycins.” What they intend exactly? The conclusions of this study should be enlarged and better explained future perspectives.
3)SPR investigations: which the equation employed to estimated kinetic parameters? Why this equation? From the shape of the sensorgrams there is a drift of the baseline…? Why? How the authors define a 1:1 ratio between proteins?
Minor
There is an error in the numeration of figures
Author Response
The paper entitled “Determination of the Protein-Protein Interactions Within Acyl Carrier Protein (MmcB) Dependent Modifications in the Biosynthesis of Mitomycin” is a very interesting paper focused on a crucial topic.
But several issues should be addressed before publication:
- the scientific message of this study resulted a bit obscure and also the last part should be rewritten.
The answer: We are sorry for the confusing description of the last conclusion part. This section has been reorganized in the current manuscript.
2) In the conclusion section where the authors write “pave the way towards the attempts to modify the chemical structures of mitomycins.” What they intend exactly? The conclusions of this study should be enlarged and better explained future perspectives.
The answer: We are sorry for the inappropriate presentations in the conclusion part. And we have revised all of the future perspectives.
3) SPR investigations: which the equation employed to estimated kinetic parameters? Why this equation? From the shape of the sensorgrams there is a drift of the baseline…? Why? How the authors define a 1:1 ratio between proteins?
The answer: This is a very good comment. The affinity of the proteins is relatively high, and it is not necessary for us to wait until they are all dissociated before proceeding to the next step in the experiment, thus causing the signal not recovery to the baseline. In fact, the data obtained at the time of partial dissociation already enough to calculate the binding and dissociation constants, and thus the affinity constants.
The kinetic evaluation procedure determines association and dissociation constants by fitting the experimental data to a 1:1 interaction model between analysis of A and ligand B:
Where ka is the association rate constant (M-1s-1), and kd is the dissociation rate constant (s-1).
The complex formation during injection is given by
and the rate of dissociation after the end of the injection is
The concentration of complex formed is measured in RU by the SPR response: if the total ligand concentration [B]0 is also expressed in RU (as the maximum analyte binding capacity), the rate equations can be written in terms of response values instead of concentrations:
Because interaction occurs at the surface of the sensor chip, analysis must be transferred laterally in the flow cell from the bulk solution to the surface before interaction with ligand can take place.
The constants reported by kinetic evaluation are determined in terms of a 1:1 interaction model. If the interaction mechanism is not 1:1 binding, the fitted curves will deviate to some extent from the experimental data and the reported constants will not be a true representation of the interaction kinetics. The apparent 1:1 binding constant can still be used for comparative studies of observed binding rates, but it is important in reporting the values to emphasize that they are empirical and not mechanistic constants.
Minor:
There is an error in the numeration of figures.
The answer: We are sorry for the inappropriate presentation. We have corrected the mistake in the current manuscript.

Round 2
Reviewer 3 Report
I agree with most of the changes made. I only have two observations that I would like the authors to respond to:
Major revision:
Figure S1-B, S2-B, S5-B, S6-B: Why was a lane added to the same figure by editing the image and the reaction was not run again?
Figure S10. Why did the authors use three different methods to reconstruct the phylogenetic tree? The reported tree from which of the three methods was it obtained? What was the model of molecular evolution that was used? Were nucleotide or amino acid sequences used? Why aren't the bootstrap values displayed in the tree?
Minor revision:
Figure S10: Change "homologies" to "homologues"
Author Response
Figure S1-B, S2-B, S5-B, S6-B: Why was a lane added to the same figure by editing the image and the reaction was not run again?
The answer: Thanks for the carefully review and kindly suggestion. In fact, our initial validations had contained the negative control sample in the gel electrophoresis analysis. Under your precise suggestion, we carried out the validation experiments again and the all of the related Figures had been updated.
Figure S10. Why did the authors use three different methods to reconstruct the phylogenetic tree? The reported tree from which of the three methods was it obtained? What was the model of molecular evolution that was used? Were nucleotide or amino acid sequences used? Why aren't the bootstrap values displayed in the tree?
Figure S10: Change "homologies" to "homologues".
The answer: We are greatly thanks for the kindly comments and suggestions. We have revised the all of the errors as we can in the manuscript. The phylogenetic analysis of MitF with its homologous had been further revised.
Finally, we would like to thanks again for your thoughtful suggestions and assistance.
Sincerely,
Dr. Qianjin KANG

Reviewer 4 Report
Even if the major of my previous issues have been satisfied the explanation related to SPR drift of base line should be added to the main text . I am not agree it is not necessary to reach baseline to perform subsequent experiment, I do not fully agree with the assumption that with another fitting equation (two sites, three sites etc) that the sensorgrams will deviate from fitting curves. To this purpose the authors should add fitting curves, residuals value and specify how the perform the fittining globally or locally as average of all experiments or kinetic from each.
Author Response
Major revision:
Even if the major of my previous issues have been satisfied the explanation related to SPR drift of base line should be added to the main text . I am not agree it is not necessary to reach baseline to perform subsequent experiment, I do not fully agree with the assumption that with another fitting equation (two sites, three sites etc) that the sensorgrams will deviate from fitting curves. To this purpose the authors should add fitting curves, residuals value and specify how the perform the fittining globally or locally as average of all experiments or kinetic from each.
The answer: We are greatly thanks for the kindly comments and suggestions. We apologize for the misunderstanding of the previous comment. According to the figure in main text, the reviewer is quite correct that there is indeed a baseline drift. We consulted Biacore engineers again and re-analyzed the experimental data. The result is shown in Fig. 1 (fit local, Chi² (RU²) 1.38e-01). Because we are using the coupling method to fix the ligands, it is possible that some floating proteins fell off during the experiment due to too much ligand coupling in the previous experiments, causing the baseline to drift. We used the coupling method to fix the ligands, it is possible that the baseline drifted due to too much ligand coupling in the previous experiments and thus some floating proteins fell off. However, the residual value, fitting model, 1:1 binding, fit local in the original data are all in accordance with the requirements.
Finally, we would like to thanks again for your thoughtful suggestions and assistance.
Sincerely,
Qianjin KANG
